# Low-Intensity Blood Flow Restriction Exercises Modulate Pain Sensitivity in Healthy Adults: A Systematic Review

**DOI:** 10.3390/healthcare11050726

**Published:** 2023-03-02

**Authors:** Stefanos Karanasios, Ioannis Lignos, Kosmas Kouvaras, Maria Moutzouri, George Gioftsos

**Affiliations:** 1Laboratory of Advanced Physiotherapy (LAdPhys), Physiotherapy Department, School of Health and Care Sciences, University of West Attica, 12243 Aigaleo, Greece; 2Physiotherapy Department, University of Patras, 20504 Patras, Greece; 3Hellenic Orthopedic Musculoskeletal Training (OMT) eDu, 11631 Athens, Greece

**Keywords:** KAATSU training, exercise, occlusion training, pain threshold, hypoalgesia

## Abstract

Low-intensity exercise with blood flow restriction (LIE-BFR) has been proposed as an effective intervention to induce hypoalgesia in both healthy individuals and patients with knee pain. Nevertheless, there is no systematic review reporting the effect of this method on pain threshold. We aimed to evaluate the following: (i) the effect of LIE-BFR on pain threshold compared to other interventions in patients or healthy individuals; and (ii) how different types of applications may influence hypoalgesic response. We included randomized controlled trials assessing the effectiveness of LIE-BFR alone or as an additive intervention compared with controls or other interventions. Pain threshold was the outcome measure. Methodological quality was assessed using the PEDro score. Six studies with 189 healthy adults were included. Five studies were rated with ‘moderate’ and ‘high’ methodological quality. Due to substantial clinical heterogeneity, quantitative synthesis could not be performed. All studies used pressure pain thresholds (PPTs) to assess pain sensitivity. LIE-BFR resulted in significant increases in PPTs compared to conventional exercise at local and remote sites 5 min post-intervention. Higher-pressure BFR results in greater exercise-induced hypoalgesia compared to lower pressure, while exercise to failure produces a similar reduction in pain sensitivity with or without BFR. Based on our findings, LIE-BFR can be an effective intervention to increase pain threshold; however, the effect depends on the exercise methodology. Further research is necessary to investigate the effectiveness of this method in reducing pain sensitivity in patients with pain symptomatology.

## 1. Introduction

Physical exercise is considered a beneficial intervention to reduce pain sensitivity in healthy individuals and patients with chronic pain conditions [1]. Clinically important changes in pain reduction are reported during or after a single bout of exercise, a phenomenon widely known as exercise-induced hypoalgesia (EIH) [2,3]. Based on experimental studies, the magnitude of EIH varies according to different factors, such as exercise parameters (i.e., type, dose, duration, and intensity), the type of noxious stimulus used for assessment (pressure, thermal, or electrical), the site of measurement (local or remote, muscle, or bone), and the timing of assessment (during or after exercise) [3,4,5,6,7]. Evidence suggests that EIH increases when the intensity of exercise is higher and over the exercising limb compared to remote sites in individuals with or without chronic pain [2,6].

During the last two decades, a new training method using low-intensity exercise with blood flow restriction (LIE-BFR) has been suggested to produce significant improvements in muscle strength, hypertrophy, and endurance in healthy individuals [8,9,10,11]. LIE-BFR is used during voluntary resistance exercises (using 20–40% of one repetition maximum [RM]) or aerobic exercises (using 50% of heart rate or VO_2_max) [12]. BFR training involves partial restriction of arterial blood flow, applying 40% to 80% of limb occlusive pressure using elastic or inflatable air cuffs of different diameters [12,13]. When air cuffs are used, external pressure is achieved with pneumatic tourniquet systems or a manual pump system [14,15]. Recently, the effectiveness of LIE-BFR has been investigated in various musculoskeletal pathologies, suggesting comparable improvements in muscle strength, hypertrophy, and function compared to traditional exercise programs [16,17,18,19]. Hence, the method has been proposed as a useful alternative in rehabilitation when high-intensity conventional exercises are contraindicated or should be avoided [20,21,22,23,24].

In addition to the positive effects on skeletal muscles, LIE-BFR has shown significant reductions in pain intensity in patients with knee problems or lateral elbow tendinopathy [16,18,23,25]. As a result, it has been suggested that the BFR component may trigger a hypoalgesic response similar to high-load resistance exercise [26,27]. Further investigations have demonstrated that LIE-BFR induces greater reductions in pressure pain thresholds compared to conventional training [28,29]. These hypoalgesic responses were partially explained by endogenous opioid and endocannabinoid system pain-modulation mechanisms [28,29]. However, other studies have supported that the addition of BFR to LIE did not provide an additional hypoalgesic response when the exercise was performed to failure [30,31]. Additionally, based on a recent randomized controlled trial (RCT), elbow flexion LIE-BFR produced a similar reduction in pain perception compared to HIE only in the exercising limb [32]. Despite growing research evidence in the current field, it remains unclear whether BFR exercises induce a significant hypoalgesic effect compared to other interventions or how different types of applications may influence the hypoalgesic response.

Although several systematic reviews and meta-analyses have investigated the effect of BFR exercises on pain intensity [21,22,33,34], there are no reviews summarizing their effect on pain sensitivity. Pain intensity describes the magnitude of experienced pain measured using subjective scales, such as the visual analogue scale, numerical rating pain scale (0–10), and other instruments during activities [35]. However, pain ratings may significantly vary due to pain sensitivity that includes complex interactions of ethnic, environmental, physical, psychosocial, and genetic factors [36]. Measuring pain sensitivity remains a complex issue in research and the evaluation of pain thresholds is most commonly used in the laboratory setting [37]. Pain threshold refers to the lowest intensity at which a given stimulus is perceived as painful, including a number of stimulus modalities, such as heat, cold, pressure, and chemical stimuli [37]. Based on the available evidence, it remains unclear if BFR exercise causes a reduction in experimentally induced pain in healthy individuals or individuals with pain. Therefore, our study intends to evaluate the effect of LIE-BFR on pain threshold compared to other interventions in healthy individuals or patients with different pathologies.

## 2. Materials and Methods

We followed the Preferred Reporting Items for Systematic Reviews and Meta-Analyses guidelines in the search strategy and reporting according to the PRISMA statement [38].

### 2.1. Search Strategy

A systematic search from inception to January 2023 was conducted using the PICOS framework (P = participants; I = interventions; C = comparison; O = outcomes, S = study design) in PubMed, CINAHL, EMBASE, PEDro, ScienceDirect, Cochrane Library, Grey literature databases and clinical trial registries [39]. Search strategy also included contact with experts in the field and manual search of the reference lists of the eligible studies. Systematic reviews were not included or assessed for quality but were examined for possible references. Moreover, internet sources were searched informally and discussions with colleagues for serendipitous discoveries were implemented to retrieve additional articles. The key terms included: “blood flow restriction” OR “ischemic training” OR “kaatsu” OR “occlusion training” OR “vascular occlusion” OR “vascular restriction” AND “pain threshold”. The full search strategy is described in Appendix A.

### 2.2. Eligibility Criteria

#### 2.2.1. Participants

Studies were considered eligible if they included healthy individuals or patients over 18 years old of any ethnicity and both sexes. Patients in eligible studies were required to present with local or widespread pain.

#### 2.2.2. Intervention

We included studies that used a standardized single bout of LIE (resistance or aerobic) incorporating restriction of blood flow alone or as an additive intervention to another type of intervention. To be considered low intensity, exercise intervention was required to include a resistance exercise using 20–40% of 1RM or an aerobic exercise at 50% of heart rate or VO_2_max [12]. Studies that included HIE-BFR (>60% of 1RM; >50% of heart rate or VO_2_max) or did not adequately report the duration and intensity of exercise were excluded.

#### 2.2.3. Comparison Groups

Studies were considered eligible if they included a control condition, placebo, sham, or any other type of intervention.

#### 2.2.4. Outcome

Outcome measures included any type of assessment of the pain threshold without restriction on the type of stimulus, e.g., pressure pain threshold, cold pressor pain tolerance, heat pain intensity, etc. Studies that used other methods of pain rating, such as subjective scales or muscle pain during exercise, were excluded.

#### 2.2.5. Study Design

We included only RCTs (parallel-group, cross-over, or pilot study designs), or quasi-randomized clinical trials if RCTs were unavailable [40], published in English. We applied no restriction on publication year. Systematic reviews, case reports, reviews, cross-sectional studies, and cohort studies were excluded from the present intervention review [41,42].

### 2.3. Study Selection and Data Extraction

After importing the search results into EndNote V.X9, two independent researchers (KK and IL) screened and selected the relevant studies using a two-step process [39]. In the first stage, each title and/or abstract was independently evaluated by the two reviewers, aiming to minimize selection bias. In the second stage, the full text for each potentially eligible study was retrieved and evaluated against the criteria for eligibility by the same independent reviewers. Any disagreement was resolved by consulting a third reviewer (SK). In parallel, the same reviewers independently extracted data from eligible studies using a standardized data extraction form. For each study, we extracted the sample size, participants’ demographic characteristics, intervention parameters, and outcomes of interest (pain threshold). In case information was missed or unclear, we communicated with the authors via email. Pain threshold is considered the minimum intensity of a stimulus (pressure, thermal, or electrical) at which an individual perceives or senses pain [43]. The available eligible studies included only the use of an algometer; therefore, we extracted only pressure pain threshold (PPT) measurements in units of kg/cm^2^.

### 2.4. Methodological Quality

Quality assessment was independently performed by two reviewers (IL and KK) using the PEDro rating scale [44]. The PEDro scale contains 11 criteria scored by a dichotomous answer (Yes/No). The first criterion is related to the study eligibility criteria and is not computed in the total score while the rest are related to the study’s internal validity and statistical reporting [44]. Based on the number of criteria satisfied (0–10), we rated the methodological quality as ‘poor’ for a score of ≤4, ‘moderate’ for a score of 5 or 6, and ‘high quality’ for a score of ≥7 [45,46,47]. Differences in the bias risk rating were discussed during a consensus meeting, and a third reviewer (GG) was involved if a consensus could not be reached.

### 2.5. Data Analysis, Summary and Synthesis of Findings

Data from the included studies were assessed for heterogeneity in a two-stage procedure. Initially, we sub-grouped the studies according to the setting, population investigated, exercise interventions, and outcomes. Then, we assessed sub-group heterogeneity using participants’ inclusion criteria, experimental and control intervention characteristics (type, equipment, volume, duration, etc.), and type of assessment (stimulus, timing, site of measurement, equipment, etc.). Because evidence for substantial clinical heterogeneity was demonstrated, we followed best-evidence synthesis to summarize the data. The risk of bias scores was considered when determining the evidence available.

## 3. Results

After the removal of duplicates, 1249 publications were identified as relevant. Screening of titles and abstracts resulted in 11 records that were considered eligible for full-text assessment. Subsequently, five studies that did not meet the eligibility criteria were excluded. Six RCTs [28,29,30,31,32,48,49] were finally included in the qualitative synthesis (Figure 1). Two of them included a parallel [32,48], while four studies followed a cross-over design [28,29,30,31].

### 3.1. Participants

A total of 189 healthy participants were included with a mean age of 24.1 years. Of the participants, 44% were women and 56% were men, and the sample sizes ranged from 12 to 60 healthy individuals. All subjects were asked to maintain normal dietary habits during participation in the experimental protocols and refrain from caffeine, alcohol, and intense exercise the day before the experimental trials. All eligible studies described some common exclusion criteria, such as serious cardiovascular diseases, venous deficiency, lymphoedema, history of heart surgery, pulmonary embolism, cancer, or thrombosis [28,29,30,31,32]. Table 1 shows the characteristics and main results of the included trials.

### 3.2. Interventions

Five studies evaluated the effectiveness of LIE-BFR alone compared to other exercise interventions [28,29,30,31,32], while only two of them included a control group [30,31]. One study compared the effectiveness of LIE-BFR between concentric and eccentric isokinetic contractions [48]. Four of the eligible studies investigated the effect of dynamic resistance exercises with BFR and one investigated the effect of low-intensity aerobic exercise with BFR [28,29,30,31,32]. Three studies used lower-limb exercises with BFR, including a leg press [29], knee extension [30], and cycling [28]. Three studies included upper-limb exercises, of which two used a dynamic elbow flexion resistance exercise [32,48] and one used an isometric handgrip contraction [31].

Three of the eligible studies used a pneumatic tourniquet system to apply the BFR condition [28,29,32]. The rest of the studies used inflatable air cuffs and determined arterial occlusive pressure with the help of a hand-held Doppler ultrasound [30,31,48]. Three studies used LIE-BFR with low occlusive pressure (40–50% AOP) [31,32,48], two studies included exercise with both low and high occlusive pressures (40% and 80% AOP, respectively) [28,29], and one trial included only high occlusive pressure (80% AOP) [30].

### 3.3. Outcome Measures

Pain sensitivity was assessed using pressure pain thresholds across all studies. Hand-held pressure algometers with a stimulation area of 1 cm diameter were used across all trials. Five studies included measurements at the exercising limb and remote sites [28,29,30,31,32] and one trial investigated the effects of exercise with BFR only at a local site (biceps brachii muscle) [48]. Five studies [28,29,30,31,32] included a follow-up measurement 5 min after exercise trials, while two studies added a follow-up assessment 24 h post-intervention [28,29]. One study examined PPTs throughout a 2-week isokinetic training program [48].

### 3.4. Methodological Quality

Table 2 shows the results of the methodological assessment based on the PEDro criteria. Out of six studies, two were rated as ‘high’, one was rated as ‘moderate’ and one was rated as ‘low quality’. Blinding therapists was not feasible in all trials due to the nature of the interventions. Four out of six studies did not ensure the blinding of the participants. One study included the blinding of outcome assessors. There was a low to no drop-out rate among the trials.

### 3.5. Effects on Pain Perception

Five studies reported significant within-group increases in PPTs after a single bout of LIE-BFR at a short-term follow-up (5 min post-intervention) [28,29,30,31,32]. Pain sensitivity was significantly decreased after a LIE-BFR using knee extension, leg press, isometric handgrip, or 20 min of cycling at both the exercising limb and distal areas of the body [28,29,30,31,32]. Based on the results of a single RCT, an elbow flexion isotonic LIE-BFR resulted in a significant increase in PPTs only at the brachialis brachii muscle and without significant changes at remote areas [32]. One RCT using elbow flexion isokinetic LIE-BFR reported no changes in PPTs at a local site of measurement throughout a 2-week training program [48].

Significantly greater increases in PPTs at local and distal sites of the body were found in favor of lower-limb LIE-BFR with 80% arterial occlusive pressure (AOP) compared to LIE-BFR with 40% AOP, HIE, and LIE alone [28,29]. Based on the same studies, LIE-BFR with 40% AOP had better results in reducing pain sensitivity than LIE alone [28,29]. Notably, the differences did not remain statistically different at the 24 h follow-up.

In contrast to previous findings, two cross-over trials found comparable decreases in PPTs between LIE-BFR and LIE, using either a resistance leg extension or a handgrip isometric exercise [30,31]. A similar hypoalgesic effect between elbow flexion LIE-BFR and HIE was found only at the biceps brachii muscle immediately after intervention [32]. No differences were found in PPTs between concentric LIE-BFR and eccentric LIE-BFR at the biceps brachii muscle using elbow flexion isokinetic exercises during a two-week training program [48].

## 4. Discussion

We analyzed six randomized controlled trials that evaluated the effectiveness of LIE-BFR compared to other exercise interventions. A total of 189 healthy subjects with a mean age of 24.1 years were included. Most eligible trials showed a ‘moderate’ or ‘high quality’ rating; however, they included small sample sizes. One study that presented a ‘poor’ methodological quality required careful consideration regarding generalizability of the study results [49]. There was substantial heterogeneity among eligible studies regarding the assessment sites and the type or volume of exercise interventions. Based on the authors’ knowledge, this is the first review to summarize the research evidence on the effectiveness of LIE-BFR on pain thresholds.

The main findings of our study suggest that various types of LIE-BFR can induce an immediate hypoalgesic effect both at local and remote sites in healthy adults. Although EIH was found to be higher after LIE-BFR compared to conventional exercise without BFR [28,29], the heightened hypoalgesic effect is possibly based on increased exercise volume due to the BFR component [30,31]. No differences were found between eccentric and concentric LIE-BFR [48].

It is well documented that HIE can been used as a pain-modulation intervention [1]; however, there is a critical question as to whether the use of an adjunct or alternative intervention such as LIE-BFR can also increase the benefits of exercise in pain conditions. Several systematic reviews have investigated the effects of LIE-BFR on pain intensity in patients with knee musculoskeletal pathologies [21,22,27,33,50]; however, their results remain inconclusive. For example, there is no evidence of further improvement in pain reduction for the use of LIE-BFR compared to conventional resistance training in patients with knee osteoarthritis [21,33]. Another review suggested significantly less knee joint pain in favor of low-intensity BFR training compared to non-BFR training in patients after ACLR [34]. These discrepancies may be attributed to the differences in the participants’ characteristics, the severity of condition and the method of BFR training [27]. In the same line, our findings suggest that the observed reductions in pain sensitivity depend on BFR methodology, such as (i) the degree of pressure (higher pressures result in greater EIH) [28,29]; (ii) the work volume (exercise to failure results in similar EIH with LIE alone) [30,31]; and (iii) the exercising limb (upper-limb exercise results in local and not remote reductions in PPTs) [32]. In another area of research, the use of cooling systems during BFR training has shown significant results in terms of safety, blood pressure changes, and point-of-care blood products (platelet-rich plasma), suggesting their potential use in cardiac rehab [51,52]. Nevertheless, evidence of the effect of such factors during BFR training on pain sensitivity is lacking and requires further investigation.

Although the effectiveness of LIE-BFR in pain reduction in patients with musculoskeletal pathologies remains unclear, a promising aspect of using the current method in a clinical setting is the potential to allow more intense exercise with less pain, especially when targeting within-session pain modulation [32,33]. Several underlying mechanisms by which LIE-BFR may induce greater EIH have been proposed [27,29]. These mechanisms involve central descending pain control pathways [53,54]; conditioned pain modulation (CPM) [55,56,57]; motor control units recruitment [4,58]; stimulation of baroreceptors [59]; metabolites production and psychological contributing factors [60,61].

Based on the results of two trials, the activation of systemic mechanisms, including CPM and stimulation of baroreceptors, was confirmed as the changes in PPTs were mediated due to the level of discomfort, increase in blood pressure, and elevated concentration of biomarkers (beta-endorphins) [29,30]. However, other RCTs have reported that EIH following different types of LIE-BFR was not mediated by similar factors [30,31,32]. Hence, the stimulation of Group III and IV afferents and high-threshold motor unit recruitment were considered the most possible mechanisms of action [30,62,63]. The various BFR exercise methodologies may potentially activate different local and systemic hypoalgesic mechanisms.

### Limitations and Future Research

The present systematic review should be interpreted in light of its limitations. First, restricting our inclusion criteria to only English-language publications may have increased the risk of missing critical information published in other languages. Second, there was a small number of well-designed studies in the current field that were also under-populated. As a result, their external validity could be compromised. Additionally, due to the substantial clinical heterogeneity of intervention protocols, a meta-analysis was not feasible. Specifically, each study included a different type of exercise (i.e., leg press, knee extension, isometric handgrip, cycling, isotonic elbow flexion, and isokinetic elbow flexion), while substantial variability was found in the workload (i.e., 75 reps or exercise to failure).

Although we intended to analyze both healthy and unhealthy individuals, there were no studies evaluating the effect of LIE-BFR on pain perception in patients with pain. Therefore, our results cannot be generalized to various pathological populations with acute or chronic pain symptomatology. Future research investigations focusing on the possible immediate hypoalgesic effects of LIE-BFR in individuals presenting with various pain conditions seem necessary. Considering that various underlying pain-modulation mechanisms have been proposed to be activated with LIE-BFR [27], more RCTs are needed to examine which exercise properties are mostly involved.

## 5. Conclusions

Based on the available data, LIE-BFR can be an effective intervention to reduce pain sensitivity at local and remote sites in healthy adults, suggesting segmental and central underlying mechanisms. However, the magnitude of the hypoalgesic effect varies depending on the exercise parameters that potentially activate different local and systemic pain-modulation mechanisms. Specifically, using LIE-BFR at the lower-limb with higher occlusive pressure (80% AOP) can induce greater hypoalgesia compared to lower pressures (40% BFR) or exercise alone. Additionally, the increased hypoalgesic effect seems to be based on the volume of exercise (exercise to failure) and the exercising limb. Further research is required to investigate the effectiveness of LIE-BFR in reducing the pain threshold in individuals with various pathological conditions.

## Figures and Tables

**Figure 1 healthcare-11-00726-f001:**
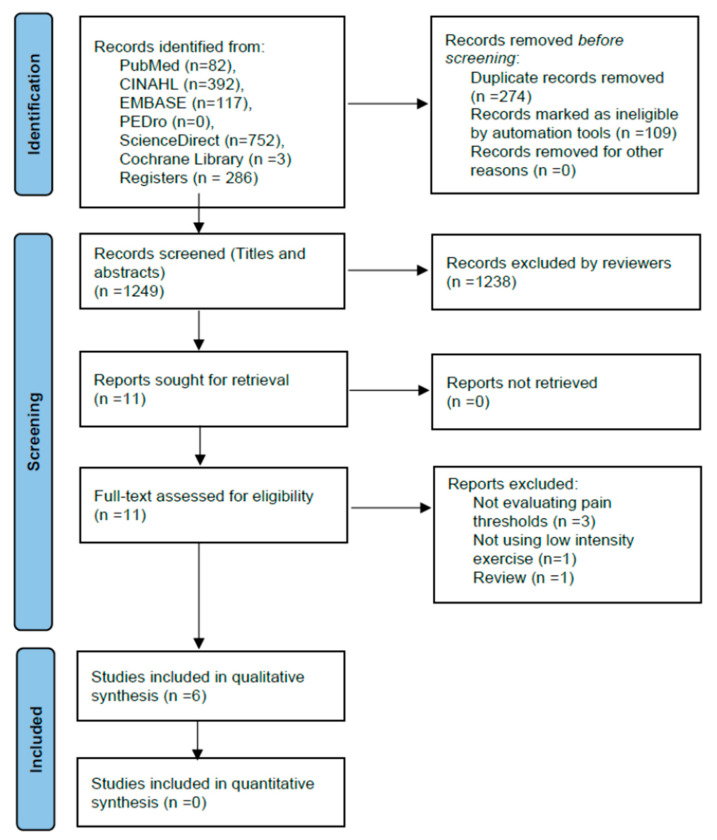
PRISMA study selection flow chart. PRISMA: preferred reporting items for systematic reviews and meta-analyses.

**Table 1 healthcare-11-00726-t001:** Included studies, demographics and results.

Study (Year)	Design	Total Sample Size N (Mean Age ± SD, Sex)	Interventions	Equipment	Follow-Up	Outcome Measures	Results
**Hill et al. (2019)** [48]	Parallel design	25 healthy individuals (25 women) Ecc LIE-BFR (n = 12; 21.7 years ± 1.0) Con LIE-BFR (n = 13; 22.0 years ± 1.6)	Unilateral isokinetic elbow flexion (120° s) (1) Ecc LIE-BFR(40% AOP) at 30% of eccentric peak torque (30-15-15-15 reps) (2) Con LIE-BFR(40% AOP) at 30% of concentric peak torque (30-15-15-15 reps)	Inflatable cuffs with a manual pump (KAATSU Master, Sato Sports Plaza, Tokyo, Japan)	Between 7 testing days	PPTs at the biceps brachii muscle	There was no significant group × testing day interaction (*p* = 0.682)
**Hughes et al. (2020)** [29]	Cross-over design	12 healthy individuals (29 ± 6 years; 10 men and 2 women)	Unilateral leg press (1) LIE using at 30% 1RM (30-15-15-15 reps) (2) LIE-BFR(40% AOP) at 30% 1RM (30-15-15-15 reps) (3) LIE-BFR(80% AOP) at 30% 1RM (30-15-15-15 reps) (4) HIE at 70% 1RM (4 sets × 10 reps)	Personalized Tourniquet system (Delfi Medical Inc, Vancouver, BC, Canada)	5 min post-exercise 24 h post-exercise	PPTs at the dominant and nondominant quadriceps, dominant biceps brachii, nondominant upper trapezius muscles	LIE-BFR (80% AOP) showed significantly higher PPTs compared to all trials (*p* < 0.05) at all measurements sites 5 min post-exercise LIE-BFR (40% AOP) showed significantly higher PPT compared to LIE (*p* < 0.05) at all measurements sites 5 min post-exercise HIE showed higher PPTs compared to LIE (*p* < 0.05) at all measurements sites 5 min post-exercise
**Hughes et al. (2021)** [28]	Cross-over design	12 healthy individuals (27 ± 6 years; 12 men)	Static bicycle X 20 min (1) LIE (2) LIE-BFR(40% AOP) at 40% VO_2_max (3) LIE-BFR(80% AOP) at 40% VO_2_max (4) HIE at 70% VO_2_max	Personalized tourniquet system (Delfi Medical Inc, Vancouver, BC, Canada)	5 min post-exercise 24 h post-exercise	PPTs at the dominant and nondominant quadriceps, dominant biceps brachii, nondominant upper trapezius muscles	PPTs were significantly increased following BFR (40% AOP) and BFR (80% AOP) compared with LIE (*p* < 0.05). BFR (80% AOP) presented higher increase in PPTs compared to BFR (40% AOP) (*p* < 0.05). BFR (80% AOP) and HI-AE presented increased PPTs in remote areas of the body.
**Karanasios et al. (2022)** [32]	Parallel design	40 healthy individuals (26.6 years ± 6.8; 17 women and 23 men)	Elbow flexion with dumbbells (1) LIE-BFR(40% AOP) at 30% RM (30-15-15-15 reps) (2) HIE at 70% RM (4 sets × 10 reps)	Personalized tourniquet system (Mad-Up Pro, France)	5 min post-exercise	PPTs at the dominant and nondominant quadriceps, biceps brachii and upper trapezius muscles	Non-significant between-group changes in PPTs at all measurement sites Statistically significant reductions between pre- and post-exercise in LIE–BFR and HIE at dominant biceps brachii
**Song et al. (2022)** [31]	Cross-over design	60 healthy individuals (21.8 years ± 3.2; 21 men, 39 women)	Isometric handgrip contraction (1) LIE-BFR (50% AOP) at 30% of max strength (4 sets × 2 min contraction) (2) LIE at 30% of max strength (4 sets × 2 min contraction) (3) control	Inflatable cuffs with a manual pump (E20, Hokanson Inc., Bellevue, WA, USA)	5 min post-exercise	PPTs at the dominant forearm and ipsilateral tibialis anterior	PPTs increased similarly in both exercise groups compared to control at a local and non-local site. Non-significant differences between exercise conditions.
**Song et al. (2022)** [30]	Cross-over design	40 healthy individuals (23.7 years ± 4.3; 18 men, 22 women)	Unilateral knee extension (1) LIE-BFR (80% AOP) at 30% RM (to failure) (2) LIE at 70% RM (to failure) (3) control	Inflatable cuffs with a manual pump (E20, Hokanson Inc., Bellevue, WA, USA)	5 min post-exercise	PPTs at the dominant forearm and ipsilateral tibialis anterior	Both exercise conditions presented greater changes in PPTs compared to control (*p* > 0.05) Non-significant differences between exercise conditions

Abbreviations: LIE-BFR, low-intensity blood flow restriction; USA, United States of America; PPTs, pressure pain thresholds, HIE, high-intensity exercise; AOP, arterial occlusive pressure; LI-AE, low-intensity aerobic exercise; HI-AE, high-intensity aerobic exercise; Ecc, eccentric; Con, concentric; RM, repetition maximum; 24 h, 24 hours; reps, repetitions.

**Table 2 healthcare-11-00726-t002:** Methodological quality assessment using the PEDro scale.

	1	2	3	4	5	6	7	8	9	10	11	Total Score
Hill et al. (2019) [48]	+	+	−	−	−	−	−	−	−	+	+	4/10
Hughes et al. (2020) [29]	+	+	+	+	−	−	−	+	+	+	+	7/10
Hughes et al. (2021) [28]	+	+	+	+	−	−	−	+	+	+	+	7/10
Karanasios et al. (2022) [32]	+	+	+	+	+	−	+	+	+	+	+	9/10
Song et al. (2022) [31]	+	+	+	−	−	−	−	+	+	+	+	6/10
Song et al. (2022) [30]	+	+	+	+	+	−	−	+	+	+	+	8/10

1. Eligibility criteria were specified; 2. subjects were randomly allocated to groups (in a cross-over study, subjects were randomly allocated an order in which treatments were received); 3. allocation was concealed; 4. the groups were similar at baseline regarding the most important prognostic indicators; 5. there was blinding of all subjects; 6. there was blinding of all therapists who administered the therapy; 7. there was blinding of all assessors who measured at least one key outcome; 8. measures of at least one key outcome were obtained from more than 85% of the subjects initially allocated to groups; 9. all subjects for whom outcome measures were available received the treatment or control condition as allocated or, where this was not the case, data for at least one key outcome were analyzed by “intention to treat”; 10. the results of between-group statistical comparisons are reported for at least one key outcome; 11. the study provides both point measures and measures of variability for at least one key outcome; Note: The first item relates to external validity and the remaining 10 items are used to calculate the total score, which ranges from 0 to 10. + Yes − No.

## Data Availability

Not applicable.

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
