# Peer review of "Low-Intensity Blood Flow Restriction Exercises Modulate Pain Sensitivity in Healthy Adults: A Systematic Review"

_healthcare, 2023, doi:10.3390/healthcare11050726_

Round 1

Reviewer 1 Report

REVIEWER Comments:

The authors have conducted a well-designed and well-executed study to investigate the effect of low-intensity exercise with blood flow restriction (LIE-BFR) on pain threshold. The study is particularly meaningful as it challenges the traditional belief that blood flow restriction exercise leads to an increase in pain. The results of this study indicate that LIE-BFR actually has a positive effect on pain threshold, which is a finding that can significantly impact the field of pain management. Congratulations on choosing a good topic. However, the authors need to check and correct some parts for the completeness of the thesis.

Introduction

1.      I noticed that the concepts of pain intensity, pain sensitivity, and pain threshold were not explained clearly in the introduction. To ensure that readers have a clear understanding of these concepts, it would be helpful to provide a brief but thorough explanation of each term.(lines 70-74)

Methods:

1.      Including the statement "This systematic review was conducted according to the PRISMA statement" adds transparency and credibility to the study. Please add recent references (revised)  

2.      The search results section could benefit from a more detailed explanation of the search criteria and methodology used to obtain the results (lines 125-130).

reference: https://www.ncbi.nlm.nih.gov/pmc/articles/PMC6045928/

Result & Discussion

1.      The discussion of the hypoalgesia effect is somewhat brief and could benefit from more detailed explanation and analysis.(based on the results of the selected studies)

Limitations and future research

1.      Including information on heterogeneity would strengthen the study and improve its overall rigor.

Conclusions:

1.      The conclusion seems clear and specific, and it effectively summarizes the main findings of the study. However, it could benefit from a more explicit explanation of the specific ways in which LIE-BFR can be effective and how it works to reduce pain sensitivity in discussion section.(lines 265-266)

Author Response

Reviewers’ comments

Responses

The authors have conducted a well-designed and well-executed study to investigate the effect of low-intensity exercise with blood flow restriction (LIE-BFR) on pain threshold. The study is particularly meaningful as it challenges the traditional belief that blood flow restriction exercise leads to an increase in pain. The results of this study indicate that LIE-BFR actually has a positive effect on pain threshold, which is a finding that can significantly impact the field of pain management. Congratulations on choosing a good topic. However, the authors need to check and correct some parts for the completeness of the thesis.

We thank the reviewer for the valuable feedback. All recommendations were considered and appropriate changes were made on the revised document.

Introduction

1.      I noticed that the concepts of pain intensity, pain sensitivity, and pain threshold were not explained clearly in the introduction. To ensure that readers have a clear understanding of these concepts, it would be helpful to provide a brief but thorough explanation of each term.(lines 70-74)

The concepts were described based on your recommendation and the paragraph was rephrased as follows

Although several systematic reviews and meta-analyses have investigated the effect of BFR exercises on pain intensity [1-4], there are no reviews summarizing their effect on pain sensitivity. Pain intensity describes the magnitude of experienced pain measured using subjective scales such as the visual analogue scale, numerical rating pain scale (0-10) and other instruments during activities [5]. However, pain ratings may significantly vary due to pain sensitivity that includes complex interactions of ethnic, environmental, physical, psychosocial, and genetic factors [6]. Although measuring pain sensitivity remains a complex issue in research, the evaluation of pain thresholds is commonly used in the laboratory setting for the current concept [7]. Pain threshold refers to the lowest intensity at which a given stimulus is perceived as painful including a number of stimulus modalities, such as heat, cold, pressure, and chemical stimuli [7]. Based on the available evidence it remains unclear if BFR exercise causes a reduction in experimentally induced pain in healthy individuals or individuals with pain. Therefore

Methods:

1.      Including the statement "This systematic review was conducted according to the PRISMA statement" adds transparency and credibility to the study. Please add recent references (revised) 

We thank you for pointing out this issue. The most recent reference was added (Page et al. 2021)

2.      The search results section could benefit from a more detailed explanation of the search criteria and methodology used to obtain the results (lines 125-130).

reference: https://www.ncbi.nlm.nih.gov/pmc/articles/PMC6045928/

Search criteria and methodology used has been rephrased and enriched with more details to provide transparency according to your recommendations. Lines 89-147.

Result & Discussion

1.      The discussion of the hypoalgesia effect is somewhat brief and could benefit from more detailed explanation and analysis.(based on the results of the selected studies)

We are very sorry for this issue. A whole section of the hypoalgesic effect was missing from the Results in the first submission (3.5 Effects on pain perception). It was added in the revised document.

Limitations and future research

1.      Including information on heterogeneity would strengthen the study and improve its overall rigor.

Information on heterogeneity was included based on reviewer’s recommendation.

Conclusions:

1.      The conclusion seems clear and specific, and it effectively summarizes the main findings of the study. However, it could benefit from a more explicit explanation of the specific ways in which LIE-BFR can be effective and how it works to reduce pain sensitivity in discussion section.(lines 265-266)

Some details were included to provide more explicit explanation of the specific ways in which LIE-BFR can be effective as recommended.

Reviewer 2 Report

Dear authors,

BFR training is representing one of the modern trends in rehabilitation to developed new methods based on exercise requiring less effort but with greater efficiency.

This systematic review is focusing on the matter of pain-reduction during BFR. The analysis is well-designed, however some corrections needs to be introduced before the acceptance:

1) Line 48-50: Also elastic tourniquets are used for BFR and it should not be overlooked, in.ex.: "Walking With Leg Blood Flow Restriction: Wide-Rigid Cuffs vs. Narrow-Elastic Bands., 2020, Frontiers in Physiology".

2) Line 82-83: The title suggests that the study regards only low-intensity BFR. Were there other than L-I training included too? If not, it cannot be stated that there was "no restriction on the type of stimulus".

3) Line 90-92: Were these the MeSH Term / Emtree? Title phrases? It needs to be specified. I would suggest adding the full search strategy in the Supplemental Material.

4) Table 1: Values of pressure used in caffs need to be added. Also, results should be not only included in this table but also described in more detail as a part of "Results" paragraph. 

5) Line 159-162: Please, specify the pressures used for flow obstruction.

6) Table 2: The quality of "Hill et al, 2019" study is questionable as we can see - it should be emphasized in the discussion.

7) Line 215 is an unnecessary repeat of the introduction. 

8) Paragraph 4.1 - Study limitations: Inclusion of only English language articles is a common practice - I would not consider it as a limitation. A small amount of well-designed studies, which in addition are under-populated is also an important issue and limitation.

9) In paragraph 4 - Discussion I am missing information on modification of BFR leading to a decrease of sensations related to this type of training. One of those is body cooling in some tools used for BFR training, which include cooling liquid inside pressure cuffs, which is supposed to also reduce pain and fatigue. I would suggest raising this in the discussion even if the studies were not included in the analysis. Here are some examples of studies with this technique "Stimulation of the Vascular Endothelium and Angiogenesis by Blood-Flow-Restricted Exercise, 2022", "Pilot Safety Study: The Use of Vasper TM, a Novel Blood Flow Restriction Exercise in Healthy Adults, 2016", "Elevation of Peripheral Blood CD34+ and Platelet Levels After Exercise With Cooling and Compression, 2021" - please read them carefully and ad some important information in the discussion.

Yours sincerely

Author Response

BFR training is representing one of the modern trends in rehabilitation to developed new methods based on exercise requiring less effort but with greater efficiency.

This systematic review is focusing on the matter of pain-reduction during BFR. The analysis is well-designed, however some corrections need to be introduced before the acceptance:

We thank the reviewer for the constructive feedback. We have strongly considered your recommendations for the resubmission of the manuscript.

1)      Line 48-50: Also elastic tourniquets are used for BFR and it should not be overlooked, in.ex.: "Walking With Leg Blood Flow Restriction: Wide-Rigid Cuffs vs. Narrow-Elastic Bands., 2020, Frontiers in Physiology".

We thank the reviewer for the suggestion. The use of elastic tourniquets was included in the current section using the proposed reference.

2)      Line 82-83: The title suggests that the study regards only low-intensity BFR. Were there other than L-I training included too? If not, it cannot be stated that there was "no restriction on the type of stimulus".

We thank the reviewer for comment. Possibly, this point was not clear in the first submission.

The type of stimulus (no restriction on the type of stimulus) refers to the outcome measure (i.e., pain thresholds) that might include hot, pressure, electrical or chemical stimulus. In the revised manuscript search strategy was rephrased including separated sections based on PICOs.

3) Line 90-92: Were these the MeSH Term / Emtree? Title phrases? It needs to be specified. I would suggest adding the full search strategy in the Supplemental Material.

We thank the reviewer for the recommendation. A search strategy was included in a supplementary material.

4) Table 1: Values of pressure used in caffs need to be added. Also, results should be not only included in this table but also described in more detail as a part of "Results" paragraph. 

We are very sorry for this issue. A whole section of the hypoalgesic effect was missing from the Results in the first submission (3.5 Effects on pain perception). It was added in the revised document.

Values of pressures in cuffs as percentage of Arterial Occlusion Pressure (AOP) was added.

5) Line 159-162: Please, specify the pressures used for flow obstruction.

We thank the reviewer for the suggestion. A paragraph including the pressures investigated was added.

6) Table 2: The quality of "Hill et al, 2019" study is questionable as we can see - it should be emphasized in the discussion.

The current point was emphasized in the first paragraph of the discussion according to your recommendation.

7) Line 215 is an unnecessary repeat of the introduction. 

The sentence was rephrased according to your recommendation.

8) Paragraph 4.1 - Study limitations: Inclusion of only English language articles is a common practice - I would not consider it as a limitation. A small amount of well-designed studies, which in addition are under-populated is also an important issue and limitation.

The paragraph was rephrased according to reviewer’s recommendation.

9) In paragraph 4 - Discussion I am missing information on modification of BFR leading to a decrease of sensations related to this type of training. One of those is body cooling in some tools used for BFR training, which include cooling liquid inside pressure cuffs, which is supposed to also reduce pain and fatigue. I would suggest raising this in the discussion even if the studies were not included in the analysis. Here are some examples of studies with this technique "Stimulation of the Vascular Endothelium and Angiogenesis by Blood-Flow-Restricted Exercise, 2022", "Pilot Safety Study: The Use of Vasper TM, a Novel Blood Flow Restriction Exercise in Healthy Adults, 2016", "Elevation of Peripheral Blood CD34+ and Platelet Levels After Exercise with Cooling and Compression, 2021" - please read them carefully and ad some important information in the discussion.

The suggested factor was included and discussed as recommended.

Round 2

Reviewer 2 Report

Dear Authors,

Thank you for the improvements that you implemented. I have no further questions or requests. Congratulations on your work.

Yours sincerely